# Community perspectives regarding brain-computer interfaces: A cross-sectional study of community-dwelling adults in the UK

**Austen El-Osta**[1]*, **Mahmoud Al Ammouri**[1], **Shujhat Khan**[2], **Sami Altalib**[1], **Manisha Karki**[1], **Eva Riboli-Sasco**[1], **Azeem Majeed**[3]

**1** Self-Care Academic Research Unit (SCARU), School of Public Health, Imperial College London, London, United Kingdom, **2** Department of Bioengineering, Imperial College London, London, United Kingdom, **3** Department of Primary Care & Public Health, School of Public Health, Imperial College London, London, United Kingdom

* a.el-osta@imperial.ac.uk

## Abstract

### Background

Brain-computer interfaces (BCIs) represent a ground-breaking advancement in neuroscience, facilitating direct communication between the brain and external devices. This technology has the potential to significantly improve the lives of individuals with neurological disorders by providing innovative solutions for rehabilitation, communication and personal autonomy. However, despite the rapid progress in BCI technology and social media discussions around Neuralink, public perceptions and ethical considerations concerning BCIs—particularly within community settings in the UK—have not been thoroughly investigated.

### Objective

The primary aim of this study was to investigate public knowledge, attitudes and perceptions regarding BCIs including ethical considerations. The study also explored whether demographic factors were related to beliefs about BCIs increasing inequalities, support for strict regulations, and perceptions of appropriate fields for BCI design, testing and utilization in healthcare.

### Methods

This cross-sectional study was conducted between 1 December 2023 and 8 March 2024. The survey included 29 structured questions covering demographics, awareness of BCIs, ethical considerations and willingness to use BCIs for various applications. The survey was distributed via the Imperial College Qualtrics platform. Participants were recruited primarily through Prolific Academic's panel and personal networks. Data analysis involved summarizing responses using frequencies and percentages, with chi-squared tests to compare groups. All data were securely stored and pseudo-anonymized to ensure confidentiality.

**Data Availability Statement:** The data underlying the results presented in the study are available in S3 File.

**Funding:** The author(s) received no specific funding for this work.

**Competing interests:** The authors have declared that no competing interests exist.

## Results

Of the 950 invited respondents, 846 participated and 806 completed the survey. The demographic profile was diverse, with most respondents aged 36–45 years (26%) balanced in gender (52% female), and predominantly identifying as White (86%). Most respondents (98%) had never used BCIs, and 65% were unaware of them prior to the survey. Preferences for BCI types varied by condition. Ethical concerns were prevalent, particularly regarding implantation risks (98%) and costs (92%). Significant associations were observed between demographic variables and perceptions of BCIs regarding inequalities, regulation and their application in healthcare. **Conclusion:** Despite strong interest in BCIs, particularly for medical applications, ethical concerns, safety and privacy issues remain significant highlighting the need for clear regulatory frameworks and ethical guidelines, as well as educational initiatives to improve public understanding and trust. Promoting public discourse and involving stakeholders including potential users, ethicists and technologists in the design process through co-design principles can help align technological development with public concerns whilst also helping developers to proactively address ethical dilemmas.

### Author summary

Brain-computer interfaces (BCIs) are emerging technologies that enable direct communication between the brain and external devices, showing promise for enhancing healthcare, especially for individuals with neurological conditions. However, despite their potential, public understanding of BCIs remains limited. This study explored community perspectives on BCIs in the UK, focusing on awareness, ethical concerns and potential applications. A survey of 806 community-dwelling adults revealed that most participants were unfamiliar with BCIs, with only a small percentage having prior experience. Respondents showed strong interest in BCIs for medical purposes, such as rehabilitation for stroke or paralysis, while expressing significant concerns about their ethical implications, including privacy, safety and the risk of exacerbating social inequalities. Participants also emphasized the importance of clear regulatory frameworks and greater public education to build trust in the technology. Notably, demographic factors such as age, gender and education influenced participants' views on the benefits and risks of BCIs. The findings highlight the need for further research to address public concerns and ensure that BCI development aligns with societal values and expectations.

## Introduction

Brain-computer interfaces (BCIs) translate electrical signals from the brain into digital commands that can be interpreted by computers. BCIs represent a rapidly evolving field that stands at the intersection of neuroscience, engineering and medicine and has the potential to transform clinical care and enhance human capabilities [1].

In the last decade, significant progress has been made in BCI technology, with multiple studies demonstrating functional restoration in neurologically impaired individuals, particularly those suffering from conditions such as stroke, trauma and degenerative disorders leading to permanent disabilities [2–4]. Examples include providing an avenue for tetraplegic individuals to communicate through neural decoding to convert speech to text as well as prosthetics

that enable movement [5–7]. BCIs can also be used in combination with functional electrical stimulation (FES) to enhance the rehabilitation process [3]. Similarly, in traumatic spinal cord injury, disruption of the neural signal from the brain to the skeletal muscles can be mitigated by connection with FES or an exoskeleton [4,8,9]. In addition to restorative capabilities, BCIs have been explored for enhancement and entertainment purposes. Examples include improving cognition, memory and strength, as well as connection with digital devices or prosthetics [10].

Beyond clinical applications, BCIs hold the potential to redefine human interaction with technology. The development of BCIs for communication in patients with locked-in syndrome—a complex medical condition presenting with quadriplegia and whole-body sensory loss [11]—illustrates the technology's ability to bridge gaps in human communication [12]. However, current limitations in speed and accuracy highlight the need for ongoing research and development to enhance the interface's reliability for broader applications, including virtual reality and neuroprosthetics [7,13]. The future of BCIs lies in refining these interfaces to achieve faster, more accurate communication and control, which would expand their clinical use and open new realms of human-computer interaction.

Whereas recent advancements in BCI technology highlight the potential for substantial therapeutic gains and enhanced human capabilities [14,15], as BCIs become increasingly sophisticated, the ethical implications of their use, particularly in non-medical contexts, warrant careful consideration [14–16]. The prospect of BCIs for cognitive and motor enhancement also raises questions about privacy, data security and the potential for exacerbating social inequalities [17]. The potential integration of BCIs with cloud computing and the use of sensitive neural data for commercial purposes also highlights an urgent need for robust ethical guidelines and regulatory frameworks to safeguard individual rights and privacy [18].

Whilst invasive BCI technology has been aimed at the treatment of clinical disorders [19], the heterogeneity in BCI technology that is currently entering FDA trials suggests that the chronic effects will vary according to the individual surgical process. Further, the use of BCIs for enhancement aimed at surpassing natural human capabilities also poses ethical concerns especially when enhancements are aimed at surpassing natural human capabilities [14,20].

Because public perceptions of BCIs are mixed [21–23], understanding public attitudes including concerns about ethical implications and societal impacts is crucial for guiding the responsible development and use of these technologies that aligns with societal values and expectations. The primary aim of this study was to investigate the knowledge, attitudes and perceptions of community-dwelling adults regarding BCIs. We also sought to uncover ethical considerations and gauge public interest in potential medical and non-medical applications. Additionally, the research explored whether demographic factors were associated with beliefs about BCIs increasing inequalities, support for strict regulations, and perceptions of appropriate fields for BCI design, testing and utilization in healthcare.

## Materials and methods

### Study design

We conducted a cross-sectional study among community-dwelling adults in the UK exploring the knowledge, attitudes, and perceptions regarding BCIs. The study adopted a quantitative methodology using an electronic survey (eSurvey).

The link to the electronic survey was active on the Imperial College Qualtrics platform between 1 December 2023 and 8 Mar 2024. The voluntary survey, which required less than 10 minutes to complete, was open and could be accessed by anyone with a link. Study information was disseminated including the Participant Information Sheet (PIS) and link to the survey. The researcher's personal and professional networks were also mobilized to respond and

further disseminate the eSurvey among potentially eligible participants. The majority of our participants were recruited via Prolific Academic's panel, an online platform where researchers can make their surveys available to participants from specific demographic backgrounds [24]. This study employed a convenience sampling approach, as this allowed data collection from individuals who were readily accessible and willing to participate. We calculated the sample size using Raosoft online software [25] with 5% margin of error, 95% confidence interval and 50% response distribution, which resulted in a recommended sample size of 385 participants.

The PIS included information regarding the study's aims, the protection of participants' personal data, their right to withdraw from the study at any time, which data were stored, where and for how long, who the investigator was, the purpose of the study and survey length. Participants were informed that this was a voluntary survey without any monetary incentives but offering the possibility to access the findings at a later stage whilst underlying the potential collective benefits of taking part in terms of helping advance knowledge in this area. Data collected were stored on a secure database at Imperial College London and only the team researchers could access the eSurvey results. All responses were pseudo-anonymised to ensure confidentiality by assigning each respondent a unique study ID. Only the participants' demographic data including age in years, gender, ethnicity, religion, residence, disability, education and employment status were recorded.

## Electronic survey

The survey comprised a structured questionnaire designed to gauge community perspectives on BCIs within the UK. It featured sections on demographic information, awareness and understanding of BCIs, ethical considerations, and willingness to use BCIs for both medical and non-medical purposes. The questionnaire was reviewed by two academic expert researchers to determine its suitability, consistency and validity. The questionnaire was also piloted with 25 eligible participants who were subsequently removed from the study. Feedback from experts was utilized to revise the questionnaire to improve design and flow and to eliminate any ambiguity. The data collected during this initial pilot was not included in the final analysis.

The survey comprised a total of 29 questions distributed over 10 pages. To enhance the survey completion rate, a maximum of three items were displayed on any one survey page. The survey (**S1 File**) was accessible using a personal computer or smartphone by following this link:

https://imperial.eu.qualtrics.com/jfe/form/SV_6m866WMXceZ9pf8

The survey was distributed online, and was accessible via various platforms including LinkedIn, Twitter/X and the Prolific Academic Panel. The survey aimed to capture a broad spectrum of opinions from a diverse cross-section of adults residing in the UK. Respondents were able to review their answers before submitting them (through a back button). The first question after the survey introduction asked participants to confirm their consent to participate in the eSurvey. Participants were then asked questions through the survey that were anonymised and not personally identifiable. The online survey technical functionality was tested before being published.

All survey items were conditional and required a response. Respondents were prompted to complete outstanding items before leaving the survey page on which the item was contained. Most items included a 'None of above/ prefer not to say' option. Relevant survey items were displayed based on previous responses (e.g., only those who have used a BCI were shown the follow-up questions about the type of BCI technology they have been using). Certain items were also populated based on previous responses. To prevent participants from completing the

survey more than once, Qualtrics XM places a browser cookie upon response submission, barring repeat attempts. Similarly, Prolific utilizes digital fingerprinting and geo-IP traps to enforce single survey completion.

## Data analysis

Quantitative data were collected using an eSurvey questionnaire administered on Qualtrics XM. Survey responses were summarised using frequencies and percentages. Chi-squared test was used to compare groups. A p-value <0.05 was considered statistically significant.

To meet the assumptions of the chi-square test for inferential analysis [26], we combined certain categories within the variables. In the Religion variable, Hinduism, Buddhism and Sikhism were merged into a single category named Dharmic faiths [27]. Similarly, in the Education variable, 'Primary school and Secondary school up to 16 years' were merged into one category. Additionally, the 'Yes with paralysis' and 'Yes without paralysis' categories in the disability-related variables were combined into 'Yes.' In addition, we also excluded 'Prefer not to say' responses and the 'non-binary' category from the Gender variable, as there was only one participant in the latter for the inferential analysis. Respondents were not excluded from the survey if they completed the items too quickly. The minimum completed survey was timed at approximately 4 minutes. Only completed questionnaires were included in the final dataset.

All analyses were performed using STATA, version 17 (StataCorp LP, College Station, TX, USA). The Checklist for Reporting Results of Internet E-Surveys (CHERRIES) was used to guide reporting [28]; **S2 File.**

## Ethical approval

The Imperial College Research Ethics Committee granted ethical clearance for the study (ICREC# 6887726). All experimental protocols were approved by Imperial College London Research Ethics Committee. All procedures performed in studies involving human participants were in accordance with the ethical standards of the institutional and/or national research committee and with the 1964 Helsinki Declaration and its later amendments or comparable ethical standards. All subjects provided consent by selecting the relevant tick box at the start of the online survey. Consent for publication is not applicable.

## Patient and public involvement

No patient was involved.

## Results

### Demographic profile of respondents

Of 950 potential respondents who were invited to participate in the survey, 846 engaged with the survey questions (89.1% participation rate), and 806 completed the survey (95.3% completion rate; 84.8% response rate). Included participants were diverse in age, gender, ethnicity and educational background (**S1 Table**). The largest proportion of respondents were in the 36–45-year (26.4%) and 26–35 (26.2%) age groups. Gender distribution was balanced with 51.6% identifying as female and 47.9% as male. Most participants identified as White (85.6%), followed by those from Asian/Asian British (7.2%) and British Black/African/Caribbean (4.6%) backgrounds. Less than a quarter (23.7%) identified as Christian, while the majority (66.7%) identified as atheists. The perceived importance of religion in life varied, with more than two-thirds (69.0%) considering it unimportant, 14.5% stating it was neither important nor unimportant, and 16.5% deeming it important. Most respondents resided in England

(84.8%). A minority (10.8%) reported having a disability without paralysis, while 23.3% had a friend or relative with a non-paralytic disability. Two-thirds (66.5%) held a college or university degree, 55.6% were employed full-time and 18.5% were in part-time employment (**S1 Table**). The main survey results are shown in **Table 1**, and the underlying data file is included **S3 File**

### Current use and knowledge about BCIs

Only a small fraction (0.7%) reported current utilization of BCIs or reported past usage (0.9%); **Table 1**. Regarding BCI familiarity, only 0.9% had comprehensive knowledge, a third (34.2%) knew of BCIs but lacked detailed knowledge, whereas most (64.9%) were unaware of BCIs before the survey. Nearly half recognized hearing aids with brain implants (46.8%) and Neuralink (46.3%), but only a few knew about Synchron (2.0%) and other BCIs (4.9%). The best-known BCI technologies were EEG (28.2%) and fMRI (28.7%). Main information sources were news outlets (44.5%), online forums (14.5%) and scientific literature (12.0%), with less input from friends, family, colleagues, healthcare professionals and advertisements.

### Preferences for BCI types to tackle impediments

Participants showed different preferences for BCI types based on various conditions. In cases of complete paralysis, 57.4% preferred invasive BCIs, 37.1% favoured non-invasive options and 5.5% showed no interest in BCIs (**Table 1**). For partial paralysis, 46.4% would choose invasive, 47.6% non-invasive and 6.0% would not consider using any BCI. When considering weakness in arms or legs, the majority (69.8%) opted for non-invasive BCIs, 19.2% for invasive and 11.0% expressed no interest. For stroke rehabilitation, 61.1% preferred non-invasive, 33.6% invasive and 5.3% showed no interest in using any of the BCI types. Preferences for BCIs also varied for conditions like speech impediments, visual impairment, bladder and bowel control issues, Parkinson's disease, ADHD, dementia, depression and anxiety, and for enhancing physical or cognitive abilities (**Table 1**).

### Interest and acceptability for current and future applications of BCIs

Over a third of respondents (38.6%) believed BCIs were currently used in healthcare and assistive technology, 19.5% in education, 10.3% in entertainment, 6.6% in workplaces, 5.5% in marketing and 19.2% in military and security sectors (**Table 1**). For BCIs in the experimental stages, 19.3% considered healthcare as the primary area for BCI development, followed by education (18.6%), military (17.6%) and entertainment (16.2%). Prospects favoured healthcare (31.9%) and education (26.8%), with lower support for military (17.5%) and entertainment (10.3%). Fewer advocated BCIs for workplace (8.7%) and marketing purposes (4.6%); **Fig 1**.

### Ethical and social concerns

The main deterrents to using BCIs were the complexity and risk of implantation (13.2%) and high costs (12.0%); **Table 1**. Other concerns included malfunction risks (11.0%), removal complexities (11.5%) and lack of historical evidence (10.6%). Lesser worries were hacking risks (6.4%), postoperative care (5.2%) and limited effectiveness (5.0%). Less prevalent concerns were personality changes (5.4%), lack of family support (1.3%) and aesthetic issues (3.2%); **Fig 2**.

Most respondents found the risks of implanting (97.6%) and removing (97.1%) BCIs significant. Concerns about malfunctions (96.8%) and effectiveness (93.4%) were high, whereas perceived high costs (91.7%), mood impacts (92.2%) and historical data deficits (90.7%) were also

**Table 1. Survey Findings.**

| | N | (%) |
|---|---|---|
| **Have you ever used a Brain-Computer Interface (BCI)?** | | |
| Yes, I am currently using one | 6 | (0.7) |
| Yes, but I no longer use it | 7 | (0.9) |
| No, never | 793 | (98.4) |
| **What type of BCI technology have you been using? † *** | | |
| Electroencephalography (EEG) | 2 | (12.5) |
| Magnetoencephalography (MEG) | 2 | (12.5) |
| Microelectrode arrays (MEAs) | 1 | (6.3) |
| Functional magnetic resonance imaging (fMRI) | 7 | (43.7) |
| Electrocorticography (ECoG), a type of intracranial electroencephalography (iEEG) | 2 | (12.5) |
| I'm not sure | 2 | (12.5) |
| **For what purpose have you been using a BCI? † *** | | |
| For medical purposes | 9 | (64.3) |
| For non-medical purposes | 5 | (35.7) |
| **How much do you know about BCIs?** | | |
| I don't know much but I have already heard of BCIs | 276 | (34.2) |
| I know a lot about BCIs | 7 | (0.9) |
| Nothing, I had never heard of BCIs before | 523 | (64.9) |
| **Which of the following BCI devices have you heard about? ‡ *** | | |
| Hearing aid with brain implant component | 163 | (46.8) |
| Neuralink (Owned by Elon Musk) | 161 | (46.3) |
| Synchron (Stentrode) | 7 | (2.0) |
| Others | 17 | (4.9) |
| **Which of the following BCI technologies have you heard about? ‡ *** | | |
| Electroencephalography (EEG) | 119 | (28.2) |
| Magnetoencephalography (MEG) | 25 | (5.9) |
| Microelectrode arrays (MEAs) | 22 | (5.2) |
| Functional magnetic resonance imaging (fMRI) | 121 | (28.7) |
| Electrocorticography (EcoG), a type of intracranial electroencephalography (iEEG) | 25 | (5.9) |
| None of the above | 110 | (26.1) |
| **Where have you heard or read about BCIs? ‡ *** | | |
| In the news (on TV, radio, online or printed newspaper) | 212 | (44.5) |
| On online forums | 69 | (14.5) |
| From friends & family | 44 | (9.2) |
| From colleagues at work | 34 | (7.1) |
| From my GP, nurse, or another healthcare professional | 23 | (4.8) |
| Through advertisement | 12 | (2.5) |
| In the scientific literature | 57 | (12.0) |
| Other | 25 | (5.4) |
| **In which of the following fields do you think BCIs are already being used? *** | | |
| Health care & assistive technology (e.g.: to support patients with paralysis or for stroke rehabilitation) | 719 | (38.6) |
| Education & learning (e.g.: to address learning disabilities) | 364 | (19.5) |
| Entertainment & recreational use (e.g.: for gaming or sports) | 191 | (10.3) |
| Workplace (e.g.: to track & improve productivity) | 122 | (6.6) |
| Marketing & commerce (e.g.: for targeted advertisement & personalised needs assessments) | 103 | (5.5) |
| Military, police & security use (e.g.: lie detection) | 357 | (19.2) |

*(Continued)*

**Table 1.** (Continued)

| | N | (%) |
|---|---|---|
| Other | 6 | (0.3) |
| **In which of the following fields do you think BCIs are currently being designed & tested (still at the experimental stage & not yet being used)?** * | | |
| Health care & assistive technology (e.g.: to support patients with paralysis, stroke rehabilitation) | 443 | (19.3) |
| Education & learning (e.g.: to address learning disabilities) | 426 | (18.6) |
| Entertainment & recreational use (e.g.: gaming, sports) | 370 | (16.2) |
| Workplace (e.g.: to track & improve productivity) | 356 | (15.6) |
| Marketing & commerce (e.g.: for targeted advertisements & personalised needs assessments) | 287 | (12.5) |
| Military, police & security use (e.g.: lie detection) | 403 | (17.6) |
| Other | 4 | (0.2) |
| **In which of the following fields do you think BCIs should be designed, tested & used?** * | | |
| Health care & assistive technology (e.g.: to support patients with paralysis, stroke rehabilitation) | 751 | (31.9) |
| Education & learning (e.g.: to address learning disabilities) | 633 | (26.8) |
| Entertainment & recreational use (e.g.: gaming, sports) | 244 | (10.3) |
| Workplace (e.g.: to track & improve productivity) | 206 | (8.7) |
| Marketing & commerce (e.g.: for targeted advertisements & personalised needs assessments) | 108 | (4.6) |
| Military, police & security use (e.g.: lie detection) | 412 | (17.5) |
| Other | 4 | (0.2) |
| **What type of BCI would you personally consider using?** * | | |
| **Complete paralysis** | | |
| Invasive | 632 | (57.4) |
| Non-invasive | 409 | (37.1) |
| I would not consider using any BCI | 61 | (5.5) |
| **Partial paralysis** | | |
| Invasive | 489 | (46.4) |
| Non-invasive | 501 | (47.6) |
| I would not consider using any BCI | 63 | (6.0) |
| **Weakness in my arms or legs** | | |
| Invasive | 174 | (19.2) |
| Non-invasive | 633 | (69.8) |
| I would not consider using any BCI | 100 | (11.0) |
| **Stroke rehabilitation** | | |
| Invasive | 329 | (33.6) |
| Non-invasive | 598 | (61.1) |
| I would not consider using any BCI | 52 | (5.3) |
| **Speech impediment** | | |
| Invasive | 221 | (24.2) |
| Non-invasive | 593 | (65.0) |
| I would not consider using any BCI | 98 | (10.8) |
| **Visual impairment** | | |
| Invasive | 330 | (34.0) |
| Non-invasive | 554 | (57.0) |
| I would not consider using any BCI | 88 | (9.0) |
| **Lack of control of my bladder** | | |
| Invasive | 312 | (32.1) |
| Non-invasive | 543 | (55.9) |
| I would not consider using any BCI | 117 | (12.0) |

(*Continued*)

**Table 1.** (Continued)

| | N | (%) |
|---|---|---|
| **Lack of control of my bowels** | | |
| Invasive | 343 | (35.0) |
| Non-invasive | 531 | (54.0) |
| I would not consider using any BCI | 108 | (11.0) |
| **Parkinson's disease** | | |
| Invasive | 525 | (49.8) |
| Non-invasive | 472 | (44.7) |
| I would not consider using any BCI | 58 | (5.5) |
| **Attention-deficit hyperactivity disorder (ADHD)** | | |
| Invasive | 99 | (11.5) |
| Non-invasive | 493 | (57.4) |
| I would not consider using any BCI | 267 | (31.1) |
| **Dementia** | | |
| Invasive | 416 | (41.1) |
| Non-invasive | 517 | (51.0) |
| I would not consider using any BCI | 80 | (7.9) |
| **Depression** | | |
| Invasive | 92 | (10.6) |
| Non-invasive | 483 | (55.9) |
| I would not consider using any BCI | 289 | (33.5) |
| **Anxiety** | | |
| Invasive | 69 | (8.1) |
| Non-invasive | 465 | (54.9) |
| I would not consider using any BCI | 313 | (37.0) |
| **To enhance my physical abilities (strength, speed)** | | |
| Invasive | 85 | (10.0) |
| Non-invasive | 305 | (35.7) |
| I would not consider using any BCI | 463 | (54.3) |
| **To enhance my cognitive abilities (memory, attention, etc)** | | |
| Invasive | 103 | (12.0) |
| Non-invasive | 390 | (45.5) |
| I would not consider using any BCI | 364 | (42.5) |
| **Would you consider getting a BCI for any other reason not listed above?** | | |
| Yes | 38 | (4.7) |
| No | 768 | (95.3) |
| **What would prevent you from getting an invasive BCI (requiring a surgical procedure)?** * | | |
| High cost of buying & maintaining the device | 541 | (12.0) |
| Stigma of having a medical device implanted | 93 | (2.1) |
| Complexity & risk of the surgical procedure to IMPLANT the BCI | 595 | (13.2) |
| Complexity & risk of the surgical procedure to REMOVE the BCI | 518 | (11.5) |
| Postoperative care (including attending further appointments) | 234 | (5.2) |
| Risks in case the BCI stops working or malfunctions | 499 | (11.0) |
| Limited effectiveness of the BCI | 226 | (5.0) |
| Concern the BCI may alter my personality | 246 | (5.4) |
| Concern the BCI may alter my mood | 203 | (4.5) |
| Lack of evidence & historical perspective regarding BCIs | 479 | (10.6) |
| Enabling further access to personal data | 169 | (3.7) |

(*Continued*)

**Table 1.** (Continued)

| | N | (%) |
|---|---|---|
| Risk of hacking of the device | 289 | (6.4) |
| General distrust in BCIs & the companies selling them | 190 | (4.2) |
| Aesthetics (it may not look good) | 146 | (3.2) |
| Lack of support or agreement from my family & friends | 58 | (1.3) |
| Lack of support or agreement from my religious community | 22 | (0.5) |
| Other | 10 | (0.2) |
| **How important are the following factors in influencing your decision to get an invasive BCI?** ** | | |
| **High cost of buying & maintaining the device** | | |
| Unimportant | 6 | (1.3) |
| Neither important nor unimportant | 33 | (7.0) |
| Important | 433 | (91.7) |
| **Stigma of having a medical device implanted** | | |
| Unimportant | 8 | (9.6) |
| Neither important nor unimportant | 24 | (28.9) |
| Important | 51 | (61.5) |
| **Complexity & risk of the surgical procedure to IMPLANT the BCI** | | |
| Unimportant | 2 | (0.4) |
| Neither important nor unimportant | 10 | (2.0) |
| Important | 501 | (97.6) |
| **Complexity & risk of the surgical procedure to REMOVE the BCI** | | |
| Unimportant | 4 | (0.9) |
| Neither important nor unimportant | 9 | (2.0) |
| Important | 434 | (97.1) |
| **Postoperative care (including attending further appointments)** | | |
| Unimportant | 99 | (47.7) |
| Neither important nor unimportant | 2 | (1.0) |
| Important | 106 | (51.2) |
| **Risks in case the BCI stops working or malfunctions** | | |
| Unimportant | 4 | (0.9) |
| Neither important nor unimportant | 10 | (2.3) |
| Important | 420 | (96.8) |
| **Limited effectiveness of the BCI** | | |
| Unimportant | 1 | (0.5) |
| Neither important nor unimportant | 12 | (6.1) |
| Important | 183 | (93.4) |
| **Concern the BCI may alter my personality** | | |
| Unimportant | 7 | (3.2) |
| Neither important nor unimportant | 10 | (4.6) |
| Important | 202 | (92.2) |
| **Concern the BCI may alter my mood** | | |
| Unimportant | 3 | (1.6) |
| Neither important nor unimportant | 12 | (6.5) |
| Important | 171 | (91.9) |
| **Lack of evidence & historical perspective regarding BCIs** | | |
| Unimportant | 4 | (1.0) |
| Neither important nor unimportant | 35 | (8.3) |
| Important | 382 | (90.7) |

(*Continued*)

**Table 1.** (Continued)

| | N | (%) |
|---|---|---|
| **Enabling further access to personal data** | | |
| Unimportant | 9 | (6.1) |
| Neither important nor unimportant | 16 | (10.9) |
| Important | 122 | (83.0) |
| **Risk of hacking of the device** | | |
| Unimportant | 100 | (40.2) |
| Neither important nor unimportant | 6 | (2.4) |
| Important | 143 | (57.4) |
| **General distrust in BCIs & the companies selling them** | | |
| Unimportant | 6 | (3.7) |
| Neither important nor unimportant | 12 | (7.3) |
| Important | 146 | (89.0) |
| **Aesthetics (it may not look good)** | | |
| Unimportant | 8 | (6.2) |
| Neither important nor unimportant | 21 | (16.3) |
| Important | 100 | (77.5) |
| **Lack of support or agreement from my family & friends** | | |
| Unimportant | 3 | (6.1) |
| Neither important nor unimportant | 9 | (18.4) |
| Important | 37 | (75.5) |
| **Lack of support or agreement from my religious community** | | |
| Unimportant | 3 | (17.7) |
| Neither important nor unimportant | 4 | (23.5) |
| Important | 10 | (58.8) |
| **Other** | | |
| Unimportant | 2 | (22.2) |
| Neither important nor unimportant | 4 | (44.5) |
| Important | 3 | (33.3) |
| **How long do you think it will be before BCIs become the new normal in the UK?** | | |
| Less than a year | 4 | (0.5) |
| 2 to 5 years | 69 | (8.6) |
| 6 to 10 years | 177 | (22.0) |
| More than 10 years | 396 | (49.1) |
| BCIs will never be used so broadly | 160 | (19.8) |
| **To what extent do you agree with the following statements regarding BCIs in general (both invasive & non-invasive)?** | | |
| **"I'm worried about the effect of BCIs being widely available to the public"** | | |
| Disagree | 216 | (26.8) |
| Neither agree nor disagree | 240 | (29.8) |
| Agree | 350 | (43.4) |
| **"I'm excited for the potential that BCIs can bring for society"** | | |
| Disagree | 141 | (17.5) |
| Neither agree nor disagree | 226 | (28.0) |
| Agree | 439 | (54.5) |
| **"People with BCIs will be more productive in their work"** | | |
| Disagree | 162 | (20.1) |
| Neither agree nor disagree | 438 | (54.3) |

(*Continued*)

**Table 1.** (Continued)

| | N | (%) |
|---|---|---|
| Agree | 206 | (25.6) |
| **"People with BCIs will feel superior to those without"** | | |
| Disagree | 291 | (36.1) |
| Neither agree nor disagree | 297 | (36.9) |
| Agree | 218 | (27.0) |
| **"BCIs will lead to an increase in inequalities"** | | |
| Disagree | 189 | (23.4) |
| Neither agree nor disagree | 294 | (36.5) |
| Agree | 323 | (40.1) |
| **"I'm worried BCIs may be implanted without proper consent"** | | |
| Disagree | 255 | (31.6) |
| Neither agree nor disagree | 177 | (22.0) |
| Agree | 374 | (46.4) |
| **"BCIs will increase stigmatisation & pathologisation of people with disability"** | | |
| Disagree | 314 | (39.0) |
| Neither agree nor disagree | 271 | (33.6) |
| Agree | 221 | (27.4) |
| **"Use of invasive BCIs in healthy patients is morally wrong"** | | |
| Disagree | 147 | (18.2) |
| Neither agree nor disagree | 257 | (31.9) |
| Agree | 402 | (49.9) |
| **"Invasive BCIs should be reserved to people with physical and/or cognitive disabilities"** | | |
| Disagree | 88 | (10.9) |
| Neither agree nor disagree | 195 | (24.2) |
| Agree | 523 | (64.9) |
| **"I support strict regulation in the development & use of BCIs even if it means technological progress"** | | |
| Disagree | 24 | (3.0) |
| Neither agree nor disagree | 113 | (14.0) |
| Agree | 669 | (83.0) |
| **"BCIs for medical purposes should be reimbursed by the NHS or insurance companies"** | | |
| Disagree | 47 | (5.8) |
| Neither agree nor disagree | 220 | (27.3) |
| Agree | 539 | (66.9) |
| **"BCIs for healthy people should be reimbursed by the government or insurance companies"** | | |
| Disagree | 458 | (56.8) |
| Neither agree nor disagree | 200 | (24.8) |
| Agree | 148 | (18.4) |
| **"BCIs should not be accessible to children (under 18 years old)"** | | |
| Disagree | 152 | (18.9) |
| Neither agree nor disagree | 269 | (33.4) |
| Agree | 385 | (47.7) |

\* = Multiple choice question (any unit of interest is number of answers and not the number of respondents)

\*\* = non-conditional responses, carried forward from participants' answers to the question: *"What would prevent you from getting an invasive BCI (requiring a surgical procedure)?"*

† = Follow-up questions for those answered *"Yes, I am currently using one"* or *"Yes, but I no longer use it"* in question *"Have you ever used a Brain-Computer Interface (BCI)?"* ‡ = Follow-up questions for those answered *"I know a lot about BCIs"* or *"I don't know much but I have already heard of BCIs"* in question *"How much do you know about BCIs?"*

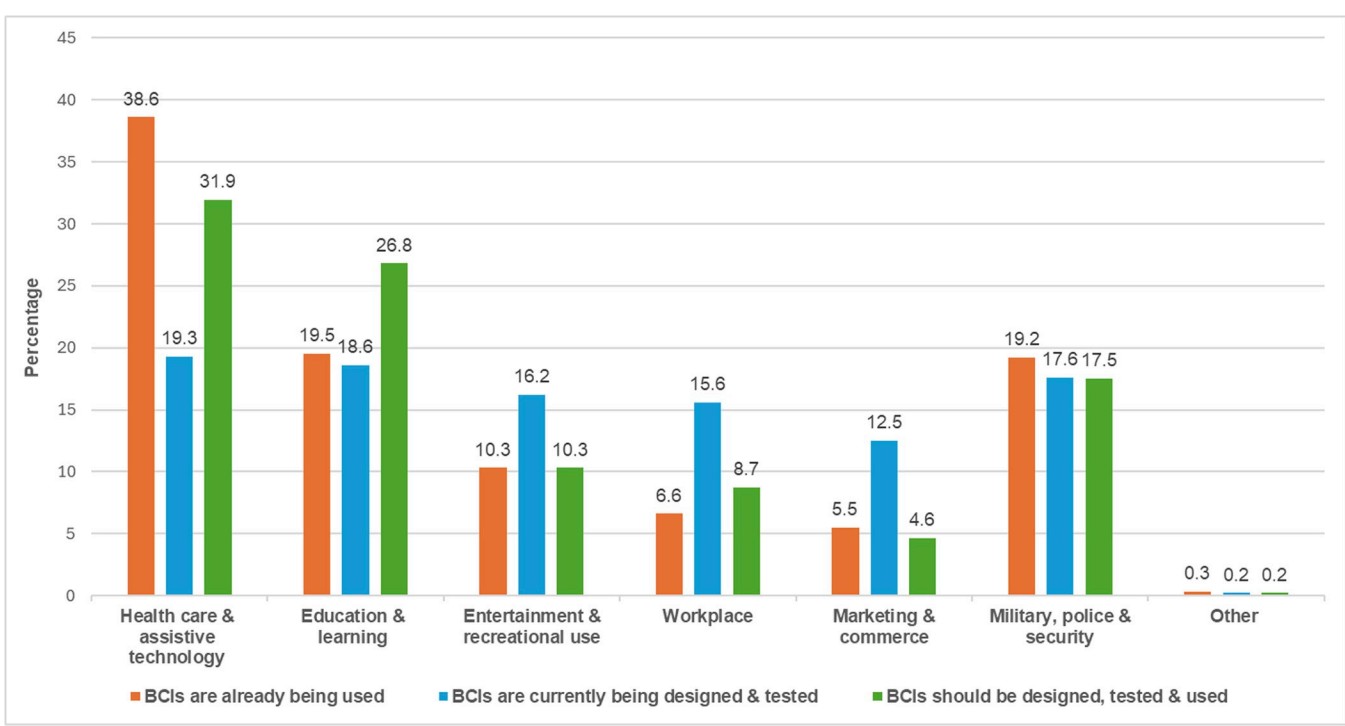

**Fig 1. Bar chart comparing public perceptions of BCIs applications in different fields across three categories: current use (orange), experimental stage (blue) and desired future use (green).**

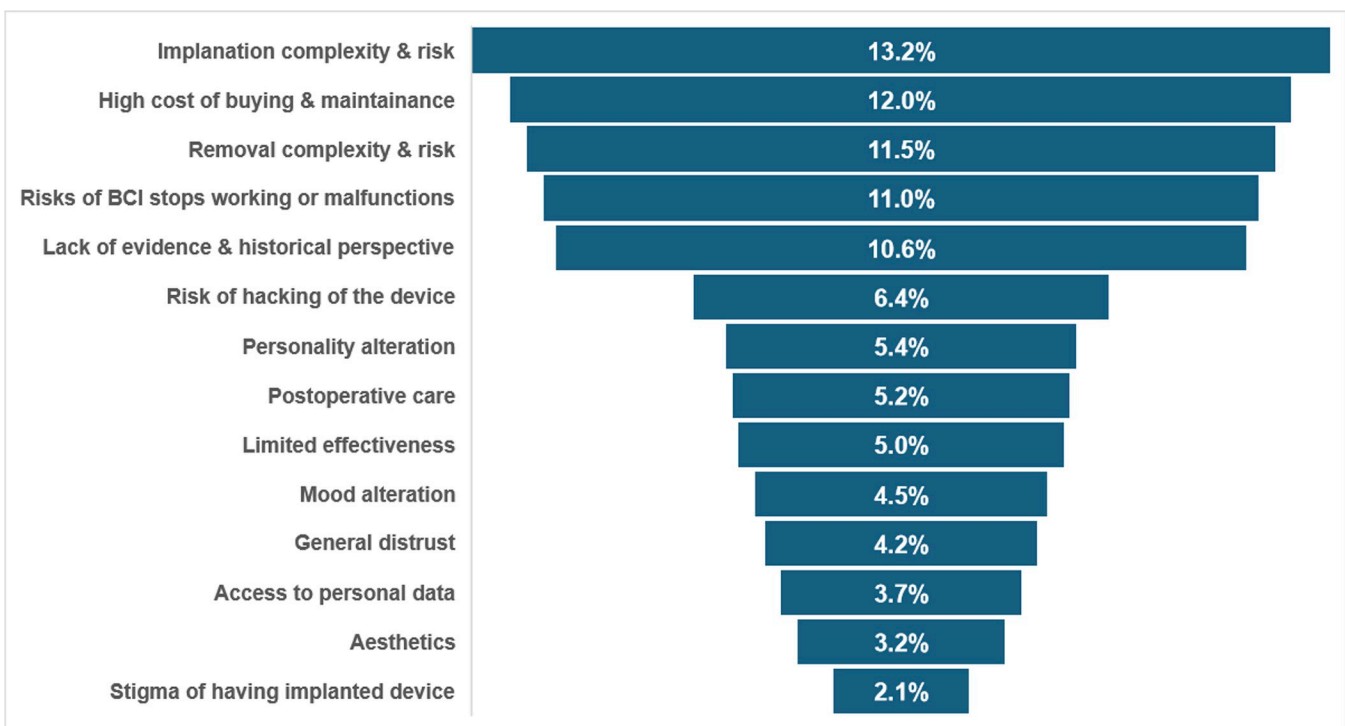

**Fig 2. Funnel chart of the perceived barriers to adopting BCIs, ranked by percentage of concern among respondents.**

notable. Hacking risks (57.4%), aesthetic (77.5%) and family support issues (75.5%) were also significant, with lower religious community support (58.8%).

Views on the impact of BCIs on productivity and inequality were mixed; 54.3% were neutral and 40.1% considered that BCIs may exacerbate inequalities. Concerns included stigma and consent, with 64.9% supporting use for disabilities, and 83.0% endorsing strict regulations. Two-thirds supported medical BCI reimbursements, while 56.8% opposed non-medical reimbursements. While 54.5% were excited about societal benefits of BCIs, 43.4% had availability concerns, and less than half (47.7%) supported BCIs for minors.

### Associations between demographics and the belief that BCIs will lead to an increase in inequalities

We found a significant association between age and the belief that BCIs will lead to an increase in inequalities ($p < 0.001$). Younger individuals (aged 26–35) demonstrated a higher tendency to agree (31.3%) compared to older age groups, particularly those aged 66 and older who showed a much lower agreement rate (2.1%); **Table 2**. There was also a significant association with gender ($p = 0.035$). Among females, 50.9% agreed that BCIs would increase inequalities, while males showed a slightly lower agreement rate of 49.1%. Neither ethnicity ($p = 0.400$) nor religion ($p = 0.005$) yielded a significant association, although 72.3% with no religion agreed that BCIs would exacerbate disparities. Conversely, education ($p = 0.019$), knowing someone with a disability ($p = 0.015$), and employment status ($p = 0.011$), were significantly associated with the belief that BCI will lead to an increase in inequalities.

### Associations between demographics and supporting strict regulation in the development and the use of BCIs even if it means technological progress

Age was significantly associated with support for strict regulation in the development and use of BCIs ($p = 0.022$), where respondents aged 18–25 showed a higher percentage of agreement (10.8%) compared to older age groups, particularly those aged 66 and older who had a lower agreement rate of 7% (**Table 3**). Gender also demonstrated a significant association ($p = 0.044$), with 53.7% of females supporting strict regulation compared to 46.3% of males. Additionally, ethnicity was significantly associated with support for regulation ($p = 0.004$), as most White respondents (87.7%) agreed with the need for strict regulation, while only 1.2% of individuals from Mixed/Multiple ethnic groups supported it. Knowing someone with a disability ($p = 0.010$), and employment status ($p = 0.004$), were also significantly associated with support for strict regulation in the development and use BCIs.

### Associations between demographics and healthcare and assistive technology field where BCIs should be designed, tested and used

Age was significantly associated with supporting designing, testing and using BCIs in the healthcare and assistive technology field ($p = 0.009$); **Table 4**. Specifically, a higher percentage of respondents aged 26–65 believed BCIs should be designed, tested and used in these fields compared to those aged 66 and older (26.5% vs. 6.7%). Religion was also significantly associated ($p = 0.008$), where a higher percentage of respondents with no religion believed BCIs should be designed, tested and used in these fields compared to those who are Christian (68.8% vs. 24.1%) or from other faiths (e.g., Dharmic faith groups at 2.5%). Furthermore, having a relative or friend with a disability was significantly associated with supporting designing, testing and using BCIs in the healthcare and assistive technology field ($p<0.001$); **Table 4**.

**Table 2. Associations between demographics and the belief that BCIs will lead to an increase in inequalities.**

| | Disagree; n (%) | Neither agree nor disagree; n (%) | Agree; n (%) | Total; n (%) | p-value |
|---|---|---|---|---|---|
| **Age** | | | | | **<0.001** |
| 18–25 | 17 (9.0) | 26 (8.8) | 57 (17.6) | 100 (12.4) | |
| 26–35 | 42 (22.2) | 68 (23.1) | 101 (31.3) | 211 (26.2) | |
| 36–45 | 50 (26.5) | 79 (26.9) | 84 (26.0) | 213 (26.4) | |
| 46–55 | 33 (17.5) | 53 (18.1) | 47 (14.6) | 133 (16.5) | |
| 56–65 | 24 (12.7) | 48 (16.3) | 27 (8.4) | 99 (12.3) | |
| 66 and Older | 23 (12.1) | 20 (6.8) | 7 (2.1) | 50 (6.2) | |
| **Gender** | | | | | **0.035** |
| Female | 85 (45.2) | 168 (57.1) | 163 (50.9) | 416 (51.9) | |
| Male | 103 (54.8) | 126 (42.9) | 157 (49.1) | 386 (48.1) | |
| **Ethnicity** | | | | | 0.400 |
| White | 166 (88.3) | 254 (86.7) | 270 (84.1) | 690 (86.0) | |
| Mixed/Multiple ethnic groups | 5 (2.7) | 3 (1.0) | 3 (0.9) | 11 (1.4) | |
| Asian/Asian British | 9 (4.8) | 20 (6.8) | 29 (9.0) | 58 (7.2) | |
| British Black/African/Caribbean | 8 (4.2) | 13 (4.4) | 16 (5.1) | 37 (4.6) | |
| Other | 0 (0.0) | 3 (1.0) | 3 (0.9) | 6 (0.8) | |
| **Religion** | | | | | **0.005** |
| Christian | 63 (34.1) | 69 (24.2) | 59 (18.6) | 191 (24.3) | |
| Dharmic (Hinduism, Buddhism or Sikhism) | 1 (0.5) | 6 (2.1) | 11 (3.5) | 18 (2.3) | |
| Islam | 7 (3.8) | 10 (3.5) | 9 (2.8) | 26 (3.3) | |
| No religion | 113 (61.1) | 196 (68.8) | 229 (72.3) | 538 (68.4) | |
| Other religion | 1 (0.5) | 4 (1.4) | 9 (2.8) | 14 (1.7) | |
| **Importance of religion in life** | | | | | 0.580 |
| Unimportant | 122 (64.6) | 207 (70.4) | 227 (70.3) | 556 (69.0) | |
| Neither important nor unimportant | 33 (17.5) | 42 (14.3) | 42 (13.0) | 117 (14.5) | |
| Important | 34 (18.1) | 45 (15.3) | 54 (16.7) | 133 (16.5) | |
| **Residence in the UK** | | | | | 0.410 |
| England | 158 (83.6) | 254 (86.4) | 272 (84.2) | 684 (84.8) | |
| Northern Ireland | 6 (3.2) | 5 (1.6) | 5 (1.6) | 16 (2.0) | |
| Scotland | 13 (6.8) | 23 (7.9) | 34 (10.5) | 70 (8.7) | |
| Wales | 12 (6.4) | 12 (4.1) | 12 (3.7) | 36 (4.5) | |
| **Disability** | | | | | 0.410 |
| No | 167 (89.8) | 258 (89.9) | 276 (86.8) | 701 (88.6) | |
| Yes (with or without paralysis) | 19 (10.2) | 29 (10.1) | 42 (13.2) | 90 (11.4) | |
| **Relative or friend with a disability** | | | | | **0.015** |
| No | 133 (73.9) | 218 (76.0) | 205 (65.7) | 556 (71.4) | |
| Yes (with or without paralysis) | 47 (26.1) | 69 (24.0) | 107 (34.3) | 223 (28.6) | |
| **Education** | | | | | **0.019** |
| Primary or secondary school up to 16 years | 22 (11.7) | 39 (13.3) | 22 (6.9) | 83 (10.4) | |
| Higher or secondary or further education (A-levels, BTEC, etc.) | 52 (27.7) | 64 (21.8) | 66 (20.7) | 182 (22.7) | |
| College or university degree | 114 (60.6) | 191 (64.9) | 231 (72.4) | 536 (66.9) | |
| **Employment status** | | | | | **0.011** |
| Employed full-time | 103 (55.4) | 158 (55.4) | 187 (58.6) | 448 (56.7) | |
| Employed part-time | 36 (19.4) | 56 (19.6) | 57 (17.9) | 149 (18.9) | |
| Retired | 22 (11.8) | 29 (10.2) | 15 (4.7) | 66 (8.4) | |
| Student | 6 (3.2) | 9 (3.2) | 27 (8.5) | 42 (5.2) | |
| Unemployed | 19 (10.2) | 33 (11.6) | 33 (10.3) | 85 (10.8) | |

**Table 3. Associations between demographics and supporting strict regulation in the development and use of BCIs even if it means technological progress.**

| | Disagree; n (%) | Neither agree nor disagree; n (%) | Agree; n (%) | Total; n (%) | p-value |
|---|---|---|---|---|---|
| **Age** | | | | | **0.022** |
| 18–25 | 7 (29.2) | 21 (18.6) | 72 (10.8) | 100 (12.4) | |
| 26–35 | 7 (29.2) | 33 (29.2) | 171 (25.6) | 211 (26.2) | |
| 36–45 | 6 (25.0) | 32 (28.2) | 175 (26.2) | 213 (26.4) | |
| 46–55 | 3 (12.4) | 16 (14.2) | 114 (17.0) | 133 (16.5) | |
| 56–65 | 1 (4.2) | 8 (7.1) | 90 (13.4) | 99 (12.3) | |
| 66 and Older | 0 (0.0) | 3 (2.7) | 47 (7.0) | 50 (6.2) | |
| **Gender** | | | | | 0.044 |
| Female | 8 (33.3) | 51 (45.1) | 357 (53.7) | 416 (51.9) | |
| Male | 16 (66.7) | 62 (54.9) | 308 (46.3) | 386 (48.1) | |
| **Ethnicity** | | | | | 0.004 |
| White | 14 (60.9) | 91 (81.3) | 585 (87.7) | 690 (86.0) | |
| Mixed/Multiple ethnic groups | 1 (4.3) | 2 (1.8) | 8 (1.2) | 11 (1.4) | |
| Asian/Asian British | 6 (26.1) | 8 (7.1) | 44 (6.6) | 58 (7.2) | |
| British Black/African/Caribbean | 2 (8.7) | 10 (8.9) | 25 (3.8) | 37 (4.6) | |
| Other | 0 (0.0) | 1 (0.9) | 5 (0.7) | 6 (0.8) | |
| **Religion** | | | | | 0.080 |
| Christian | 2 (9.5) | 28 (25.7) | 161 (24.5) | 191 (24.3) | |
| Dharmic (Hinduism, Buddhism or Sikhism) | 1 (4.8) | 2 (1.8) | 15 (2.3) | 18 (2.3) | |
| Islam | 3 (14.3) | 6 (5.5) | 17 (2.5) | 26 (3.2) | |
| No religion | 15 (71.4) | 72 (66.1) | 451 (68.7) | 538 (68.4) | |
| Other religion | 0 (0.0) | 1 (0.9) | 13 (2.0) | 14 (1.8) | |
| **Importance of religion in life** | | | | | 0.500 |
| Unimportant | 14 (58.4) | 77 (68.1) | 465 (69.5) | 556 (69.0) | |
| Neither important nor unimportant | 5 (20.8) | 13 (11.5) | 99 (14.8) | 117 (14.5) | |
| Important | 5 (20.8) | 23 (20.4) | 105 (15.7) | 133 (16.5) | |
| **Residence in the UK** | | | | | 0.500 |
| England | 20 (83.3) | 100 (89.3) | 564 (84.2) | 684 (84.8) | |
| Northern Ireland | 1 (4.2) | 3 (2.7) | 12 (1.8) | 16 (2.0) | |
| Scotland | 1 (4.2) | 6 (5.3) | 63 (9.4) | 70 (8.7) | |
| Wales | 2 (8.3) | 3 (2.7) | 31 (4.6) | 36 (4.5) | |
| **Disability** | | | | | 0.090 |
| No | 24 (100) | 101 (91.8) | 576 (87.7) | 701 (88.6) | |
| Yes (with or without paralysis) | 0 (0.0) | 9 (8.2) | 81 (12.3) | 90 (11.4) | |
| **Relative or friend with a disability** | | | | | 0.010 |
| No | 22 (91.7) | 86 (78.9) | 448 (69.3) | 556 (71.4) | |
| Yes (with or without paralysis) | 2 (8.3) | 23 (21.1) | 198 (30.7) | 223 (28.6) | |
| **Education** | | | | | 0.250 |
| Primary or secondary school up to 16 years | 1 (4.2) | 16 (14.2) | 66 (9.9) | 83 (10.4) | |
| Higher or secondary or further education (A-levels, BTEC, etc.) | 4 (16.7) | 30 (26.6) | 148 (22.3) | 182 (22.7) | |
| College or university degree | 19 (79.2) | 67 (59.2) | 450 (67.8) | 536 (66.9) | |
| **Employment status** | | | | | 0.004 |
| Employed full-time | 15 (62.5) | 70 (63.6) | 363 (55.3) | 448 (56.7) | |
| Employed part-time | 1 (4.2) | 17 (15.5) | 131 (20.0) | 149 (18.9) | |
| Retired | 1 (4.2) | 3 (2.7) | 62 (9.5) | 66 (8.4) | |
| Student | 5 (20.8) | 7 (6.4) | 30 (4.6) | 42 (5.2) | |
| Unemployed | 2 (8.2) | 13 (11.8) | 70 (10.6) | 85 (10.8) | |

**Table 4. Associations between demographics and health care and assistive technology field where BCIs should be designed, tested and used.**

| | No; n (%) | Yes; n (%) | Total; n (%) | p-value |
|---|---|---|---|---|
| **Age** | | | | 0.009 |
| 18–25 | 15 (27.3) | 85 (11.3) | 100 (12.4) | |
| 26–35 | 12 (21.8) | 199 (26.5) | 211 (26.2) | |
| 36–45 | 15 (27.3) | 198 (26.4) | 213 (26.4) | |
| 46–55 | 8 (14.5) | 125 (16.6) | 133 (16.5) | |
| 56–65 | 5 (9.1) | 94 (12.5) | 99 (12.3) | |
| 66 and Older | 0 (0.0) | 50 (6.7) | 50 (6.2) | |
| **Gender** | | | | 0.070 |
| Female | 22 (40.0) | 394 (52.7) | 416 (51.9) | |
| Male | 33 (60.0) | 353 (47.3) | 386 (48.1) | |
| **Ethnicity** | | | | 0.140 |
| White | 41 (74.6) | 649 (86.9) | 690 (86.0) | |
| Mixed/Multiple ethnic groups | 1 (1.8) | 10 (1.3) | 11 (1.4) | |
| Asian/Asian British | 7 (12.7) | 51 (6.8) | 58 (7.2) | |
| British Black/African/Caribbean | 5 (9.1) | 32 (4.3) | 37 (4.6) | |
| Other | 1 (1.8) | 5 (0.7) | 6 (0.8) | |
| **Religion** | | | | 0.008 |
| Christian | 14 (26.4) | 177 (24.1) | 191 (24.3) | |
| Dharmic (Hinduism, Buddhism or Sikhism) | 0 (0) | 18 (2.5) | 18 (2.3) | |
| Islam | 6 (11.3) | 20 (2.7) | 26 (3.2) | |
| No religion | 33 (62.3) | 505 (68.8) | 538 (68.4) | |
| Other religion | 0 (0.0) | 14 (1.9) | 14 (1.8) | |
| **Importance of religion in life** | | | | 0.060 |
| Unimportant | 35 (63.6) | 521 (69.4) | 556 (69.0) | |
| Neither important nor unimportant | 5 (9.1) | 112 (14.9) | 117 (14.5) | |
| Important | 15 (27.3) | 118 (15.7) | 133 (16.5) | |
| **Residence in the UK** | | | | 0.690 |
| England | 48 (87.3) | 636 (84.7) | 684 (84.8) | |
| Northern Ireland | 0 (0.0) | 16 (2.1) | 16 (2.0) | |
| Scotland | 4 (7.3) | 66 (8.8) | 70 (8.7) | |
| Wales | 3 (5.4) | 33 (4.4) | 36 (4.5) | |
| **Disability** | | | | 0.160 |
| No | 51 (94.4) | 650 (88.2) | 701 (88.6) | |
| Yes (with or without paralysis) | 3 (5.6) | 87 (11.8) | 90 (11.4) | |
| **Relative or friend with a disability** | | | | <0.001 |
| No | 51 (92.7) | 505 (69.7) | 556 (71.4) | |
| Yes (with or without paralysis) | 4 (7.3) | 219 (30.3) | 223 (28.6) | |
| **Education** | | | | 0.770 |
| Primary or secondary school up to 16 years | 7 (12.7) | 76 (10.2) | 83 (10.4) | |
| Higher or secondary or further education (A-levels, BTEC, etc.) | 11 (20.0) | 171 (22.9) | 182 (22.7) | |
| College or university degree | 37 (67.3) | 499 (66.9) | 536 (66.9) | |
| **Employment status** | | | | 0.110 |
| Employed full-time | 36 (70.6) | 412 (55.8) | 448 (56.7) | |
| Employed part-time | 9 (17.6) | 140 (18.9) | 149 (18.9) | |
| Retired | 0 (0.0) | 66 (8.9) | 66 (8.4) | |
| Student | 1 (2.0) | 41 (5.6) | 42 (5.2) | |
| Unemployed | 5 (9.8) | 80 (10.8) | 85 (10.8) | |

## Discussion

### Summary of principal findings

Our comprehensive study on community perspectives regarding BCIs in the UK highlights a significant gap between the potential of BCI technology and public awareness or engagement with these systems. Despite nearly all respondents (98.4%) indicating they had never used a BCI, there was a notable curiosity and openness to their medical applications, particularly in aiding individuals with disabilities or neurological conditions. Additionally, our study identified significant associations between various (a) demographic factors (age, gender, ethnicity, education level, employment status), or (b) having a friend or relative with a disability, and (i) inequalities, (ii) BCI regulation, and (iii) their application in healthcare.

Respondents distinctly favoured the medical and rehabilitative applications of BCIs over non-medical uses. This was unsurprising as it aligned with a broader societal value placed on technology's role in enhancing healthcare and quality of life for individuals facing physical challenges. For instance, the prospect of using BCIs in stroke rehabilitation or to assist those with complete or partial paralysis was met with considerable approval, highlighting the community's recognition of BCIs as a beneficial innovation in medical technology [29]. This optimistic viewpoint was tempered by significant reservations about ethical, privacy and safety implications of BCIs, especially in non-medical contexts. Participants expressed concerns largely regarding the potential for data security risks and the long-term societal impacts of widespread BCI adoption. Such apprehensions point to an urgent need for clear regulatory frameworks and ethical guidelines to navigate the future development and implementation of BCI technologies given the rising rhetoric about Neuralink as publicised by Elon Musk [30] with various posts also appearing on X (formerly Twitter). Another critical finding was the public's call for increased education and engagement around BCIs since the lack of familiarity with BCI technology suggests a disconnect between scientific advancements and public knowledge. Thus, whilst our sample of respondents from a cross-section of the UK community showed a cautious optimism toward BCIs, particularly in their ability to address complex medical needs, there was a clear mandate for addressing ethical concerns and enhancing public understanding of these technologies.

The mixed perceptions of BCIs among the public can be attributed to various factors, including demographic variables. Research indicates that younger individuals tend to exhibit greater openness to technological innovations, while older populations may harbour scepticism due to concerns about safety and ethical implications. For example, older and less educated individuals reported being more likely to reject invasive neurotechnologies, perceiving them as dangerous or unnatural [22]. Moreover, education plays a crucial role in shaping perceptions; individuals with higher educational attainment often demonstrate a better understanding of technology, leading to more favourable attitudes towards BCIs [31] whereas, by contrast, those with limited exposure may view BCIs as invasive or risky. Similarly, pre-existing attitudes significantly affect how different demographic groups perceive BCIs, with less educated individuals often expressing greater concerns about the risks associated with these technologies [32].

### Comparison with existing literature

Our research findings are consistent with existing literature on public attitudes toward BCIs technology. A survey conducted by The Pew Research Centre among U.S. adults revealed significant insights [33] as 61% of respondents indicated they had no prior exposure to the idea of implanted computer chips in the brain. Additionally, a substantial 68% expressed concerns

regarding such implants. Many U.S adults view these technological advancements as morally unacceptable, although a considerable number remain uncertain about their ethical implications. Another survey highlighted that 77% of the U.S. population supported the use of brain chips to assist individuals with paralysis, but 57% believed that the integration of brain chips could worsen socioeconomic disparities. These findings highlight the complexity of public sentiment surrounding BCI technology and ethicality [23].

Most apprehensions about BCIs expressed in our study centred on the complexity and risks associated with the surgical procedures required for implantation and removal, aligning with the conclusions of Sattler et al., who noted similar sentiments regarding the risks involved in BCI procedures [22]. As with any surgical invasive procedure, implantation carries the risk of infection, haemorrhage and iatrogenic injury [34] although chronic implantation of BCIs has yet to be studied in detail. Few studies found that in individuals with chronic brain implants, the concurrent formation of glial scar tissue at the insertion site accompanied by micromotion of implants significantly diminishes the longevity of reliably recorded signals and reduces their quality over time [35,36]. As such, the risks associated are not clear, highlighting the necessity for monitoring the wellbeing of chronic implant patients, while also facilitating the assessment of long-term efficacy. Additionally, Intravascular implantation [37] carries a distinct set of risks which can result in a stroke or haemorrhage [34] and it is again not yet clear whether the risk of thrombosis is worrisome or whether patients may need to take antiplatelet/antithrombotic post-procedure as other neuro-interventional procedures [38].

Furthermore, aesthetic concerns may influence the decision whether to adopt a BCI device [39]. Our findings resonate with another study that identified concerns about the cosmetic visibility of BCIs, particularly among individuals with physically stigmatizing conditions like paralysis from spinal cord injury [40]. Such cosmetic barriers could significantly hinder the adoption of these devices, despite their potential for substantial functional restoration.

Our findings from the UK align with a growing body of literature that situates BCIs at the intersection of medical innovation, ethical scrutiny and privacy concerns. Consistent with previous studies, we observed a strong interest in the medical applications of BCIs, particularly in assisting patients who have lost motor and sensory functions due to stroke or spinal cord injuries [41,42]. Our study's emphasis on ethical and privacy issues reflects cautionary perspectives raised by other researchers [14,43] who critically examined the societal implications and moral responsibilities associated with BCI advancements. This convergence of interests highlights the need for a comprehensive approach to BCI development that balances innovation with ethical considerations and public trust.

An important concern from participants is the risk of devices getting hacked. Comparable to the risks of hacking associated with other medical implants such as cardiac defibrillators [44], BCIs are associated with potential risks of neuro-hacking as well [45–47]. This includes wireless manipulation of device settings that can produce potentially harmful brain stimulation affecting an individual's cognitive, or physical characteristics, alongside the interception of signals from a brain implant to unveil sensitive personal data [46,47]. These valid concerns highlight the need for balance between innovation and ethical governance—a theme that is recurrent in both our findings and the broader academic discourse [48]—and makes the case for developing and implementing agile and resilient regulatory frameworks to ensure the privacy and safety of BCI devices. Our finding that the perceived the high costs associated with invasive BCI devices could exacerbate existing inequalities also aligns with previous studies indicating a growing concern about socio-economic disparities related to accessing BCI technology and this raises ethical concerns about equity in healthcare and technology access [49].

Our study findings notably diverge in the detailed examination of public attitudes toward non-medical uses of BCIs emphasising various apprehensions and encroachments on ethical

boundaries perceived by the public. These reservations corroborate and enrich the ongoing debate around the commercialization and recreational use of BCIs as areas that are likely to be increasingly scrutinized as technology advances [17]. This is particularly relevant in the cases of enhancement whereby BCIs can be used to improve physical characteristics, which may be seen as a form of neural doping that offers the potential to surpass natural physical limits, particularly when combined with stimulatory capabilities [50].

## Implications for research

The strong support for the medical application of BCIs, especially in rehabilitation for individuals with disabilities, highlights a clear direction for prioritizing research and development efforts [51] and for funding agencies and research institutions to allocate resources to these applications to maximize societal benefits. As BCIs evolve, comprehensive policies addressing potential socio-economic disparities should be developed in consultation with diverse stakeholders to reflect a wide array of societal values and concerns.

Ethical reflection and dialogue on issues like privacy and data security are crucial as society navigates the complexities of BCI technology [52–54] and self-driven healthcare solutions to promote health, wellbeing and the lived experience. A participatory approach can help align technological advancements with public expectations and to help build trust and transparency. Given public concerns about data security and misuse, ethical considerations must be prioritized in BCI design and implementation whereas robust regulatory frameworks can help protect individual privacy and ensure responsible use of BCIs. Again, the adoption of co-design principles including involvement by potential users, ethicists and technologists in the design process can bridge the gap between technological development and public concerns. However, while our research indicates a desire for increased awareness about BCIs, simply raising awareness alone may not alleviate public concerns. Many respondents understood the potential benefits of BCIs in medical contexts but remained apprehensive about ethical implications and safety risks suggesting a gap between BCI technology's potential and public confidence in its application. To bridge this gap, educational programs should provide comprehensive insights into BCI technology, addressing medical applications alongside ethical considerations, privacy issues and technological limitations. Ongoing dialogue between researchers and the public can demystify BCIs while addressing fears rooted in misunderstandings or misconceptions, whilst helping developers to proactively address ethical dilemmas.

## Limitations

To our knowledge, this is the first study to comprehensively examine community perspectives on BCIs in the UK. Although the findings from our sample offer a robust reflection of societal views towards BCIs in a healthcare context, there are a number limitations that hinger generalisability of our findings including the use of a convenience sample which may introduce selection bias. Another limitation stems from the sample's non-representative religious demography as the study sample largely included respondents who indicated they had no religion (67%) which is nearly double the national average reported in the National Census figures from 2021 in England and Wales [55] where this demographic stood at 37%. Additionally, despite the high response rate to the study, we acknowledge that nonresponse bias may still pose a concern, as the characteristics and opinions of those who did not participate could differ from those who did, potentially skewing our study findings. We acknowledge also that the cross-sectional design of our study limits our ability to establish causality between variables and hinders our capacity to track changes in public opinion over time. Further, our reliance on self-reported data may also introduce bias, as participants' responses could be influenced

by social desirability or a lack of understanding of the complex technical aspects of BCIs. Finally, although our sample is diverse, it predominantly comprised of 806 individuals with a higher than average proportion of respondents that attained a university degree, again potentially skewing the perceived enthusiasm and concerns regarding BCIs.

## Conclusion

Our study expands the existing literature on BCIs by offering a focused lens on UK community perspectives. Despite a strong interest in BCIs, particularly for medical applications, ethical concerns, privacy issues and the perceived safety of BCIs are highlighted as well as the concerns associated with these technologies potentially exacerbating inequalities. These valid observations necessitate the need for clear regulatory frameworks and ethical guidelines, as well as educational initiatives to promote public understanding and trust. Involving a wide mix of stakeholders including potential users, ethicists and technologists in the design process using co-design principles can help align technological development with public concerns whilst also helping developers to proactively address ethical dilemmas.

## Supporting information

**S1 File. Survey (main data collection tool).**
(DOCX)

**S2 File. Checklist for Reporting Results of Internet E-Surveys (CHERRIES) was used to guide reporting.**
(DOCX)

**S3 File. Survey responses data file.**
(XLS)

**S1 Table. Respondent characteristics.**
(DOCX)

## Acknowledgments

The authors wish to thank Aos Alaa for his support in survey beta testing and development. Austen El-Osta, Mahmoud Al Ammouri, Sami Altalib and Azeem Majeed are supported by the National Institute for Health and Care Research (NIHR) Applied Research Collaboration (ARC) Northwest London. The views expressed are those of the authors and not necessarily those of the NHS or the NIHR or the Department of Health and Social Care.

## Author Contributions

**Conceptualization:** Austen El-Osta, Shujhat Khan.

**Data curation:** Manisha Karki.

**Formal analysis:** Mahmoud Al Ammouri, Sami Altalib.

**Methodology:** Shujhat Khan, Sami Altalib.

**Supervision:** Eva Riboli-Sasco, Azeem Majeed.

**Validation:** Austen El-Osta, Shujhat Khan.

**Writing – original draft:** Austen El-Osta, Mahmoud Al Ammouri.

**Writing – review & editing:** Austen El-Osta, Shujhat Khan, Sami Altalib, Manisha Karki, Eva Riboli-Sasco, Azeem Majeed.

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
