## [Decision Letter · Decision Letter 0]

1 Aug 2024

PDIG-D-24-00179

What are community perspectives regarding brain-computer interfaces? A cross-sectional study of community-dwelling adults in the UK

PLOS Digital Health

Dear Dr. El-Osta,

Thank you for submitting your manuscript to PLOS Digital Health. After careful consideration, we feel that it has merit but does not fully meet PLOS Digital Health's publication criteria as it currently stands. Therefore, we invite you to submit a revised version of the manuscript that clearly addresses the points raised during the review process.

Please submit your revised manuscript within 60 days Sep 30 2024 11:59PM. If you will need more time than this to complete your revisions, please reply to this message or contact the journal office at digitalhealth@plos.org. Please include the following items when submitting your revised manuscript:

We look forward to receiving your revised manuscript.

Kind regards,

Avinash Singh, PhD

Academic Editor

PLOS Digital Health

Journal Requirements:

1. We do not publish any copyright or trademark symbols that usually accompany proprietary names, eg (R), (C), or TM (e.g. next to drug or reagent names). Please remove all instances of trademark/copyright symbols throughout the text, including ® on page 5.

Additional Editor Comments (if provided):

Reviewers' comments:

Reviewer's Responses to Questions

**Comments to the Author**

1. Does this manuscript meet PLOS Digital Health’s publication criteria? Is the manuscript technically sound, and do the data support the conclusions? The manuscript must describe methodologically and ethically rigorous research with conclusions that are appropriately drawn based on the data presented.

Reviewer #1: Yes

Reviewer #2: Partly

Reviewer #3: Yes

2. Has the statistical analysis been performed appropriately and rigorously?

Reviewer #1: Yes

Reviewer #2: I don't know

Reviewer #3: Yes

3. Have the authors made all data underlying the findings in their manuscript fully available (please refer to the Data Availability Statement at the start of the manuscript PDF file)?

Reviewer #1: No

Reviewer #2: No

Reviewer #3: Yes

4. Is the manuscript presented in an intelligible fashion and written in standard English?

Reviewer #1: Yes

Reviewer #2: No

Reviewer #3: Yes

5. Review Comments to the Author

Reviewer #1: The paper presents a comprehensive survey on public perspectives regarding Brain-Computer Interfaces (BCIs) among community-dwelling adults in the UK. The paper is well-written and easy to understand. Here are comments and suggestions to further improve the paper:

1. Discussion on Non-Response Bias: The paper reports a high response rate, but there is little discussion on the potential for non-response bias. 

2. Validity of the Results: Did the authors ensure that the results of a survey accurately reflect the perspectives of the targeted population in some ways?

3. Figures and Graphical Representations: Adding figures to visually represent the results, such as bar graphs of demographic breakdowns or pie charts showing levels of support for different applications of BCIs, would make the data more accessible and easier to interpret at a glance.

Reviewer #2: The title is clear but why using question format.

The abstract is concise and provides a clear summary of the study’s objectives, methods, and key findings. However, including specific numerical results and a brief mention of the main conclusions is needed.

The study design is robust. However, the inclusion of a pilot test for the survey instrument should be mentioned to validate the questionnaire’s reliability and comprehensibility.

The results section is comprehensive. However, the narrative could benefit from a more explicit linkage of the results to the study’s aims and research questions.

The discussion needs a deeper exploration of the reasons behind the public’s mixed perceptions and the potential impact of demographic variables on attitudes towards BCIs.

The conclusion is missing specific recommendations for policymakers, researchers, and practitioners about its practical relevance.

The sampling method is not clearly described. It is unclear whether random, convenience, or purposive sampling was used.

The sample size of 806 is mentioned but there is no justification for this number. A power calculation to determine the appropriate sample size is missing.

There is no information on the validity and reliability of the questionnaire used.

The method of administration (online, face-to-face, etc.) is not specified.

The response rate is not reported, which is crucial for assessing the representativeness of the sample.

Ethical approval and informed consent procedures are not mentioned, raising concerns about the ethical rigor of the study.

The results section is primarily descriptive with limited inferential statistics. The study does not seem to leverage advanced statistical analyses to draw more robust conclusions.

There is a lack of clarity in the presentation of demographic data and their implications on the study’s findings.

Tables and figures are poorly integrated into the text and do not enhance the understanding of the results.

The reported percentages in the tables do not always add up to 100%, indicating potential errors in data presentation or analysis.

The discussion is superficial and does not critically engage with the findings.

There is a lack of comparison with previous studies, which limits the study’s contribution to the existing body of knowledge.

https://doi.org/10.1371/journal.pone.0296884

https://doi.org/10.1186/s12910-024-01062-8

Potential biases and limitations are either underexplored or ignored altogether.

The implications of the findings for future research, policy, and practice are not adequately discussed.

The conclusion is weak and does not effectively summarize the main findings or their significance.

Recommendations for future research are vague and lack specificity.

The overall structure of the paper lacks coherence, making it difficult to follow the logical flow of arguments.

Language and grammar issues are present throughout the manuscript, which detracts from the professionalism of the paper.

There is a noticeable absence of critical engagement with the literature and a lack of depth in the analysis.

The study's contribution to the field is minimal due to its methodological flaws and superficial treatment of the topic.

Reviewer #3: This is a very investing study and I was very happy to see that the authors directly engage with some of the concerns that the public might have about BCI and related applications. I think that the conclusions reached in the study are very interesting and show a direct engagement with these issues. I recommend publication and only have smaller suggestions. 

First, I wonder if the authors could say a bit more on the consequences of their findings for research and perhaps their research: what should be done considering the reservations about the ethical, privacy and safety implications? Could a path towards co-design or ethically-informed design be developed? 

Secondly and relatedly, the paper seems to imply that the identified issues are due to and will be resolved by more awareness and knowledge about the technology. I am not so sure about this, as some of the concerns seem to be aligned with how the technology works or at least could be developed. I wonder if the authors could say more about the idea of a gap between the potential of the technology and public awareness – to me it seems that the public was actually mostly aware of the potential of the technology and still concerned.

6. PLOS authors have the option to publish the peer review history of their article (what does this mean?). If published, this will include your full peer review and any attached files.

**Do you want your identity to be public for this peer review?** For information about this choice, including consent withdrawal, please see our Privacy Policy.

Reviewer #1: No

Reviewer #2: No

Reviewer #3: No

---

## [Decision Letter · Decision Letter 1]

20 Dec 2024

Community perspectives regarding brain-computer interfaces: a cross-sectional study of community-dwelling adults in the UK

PDIG-D-24-00179R1

Dear Dr. El-Osta,

We are pleased to inform you that your manuscript 'Community perspectives regarding brain-computer interfaces: a cross-sectional study of community-dwelling adults in the UK' has been provisionally accepted for publication in PLOS Digital Health.

Best regards,

Baki Kocaballi

Section Editor

PLOS Digital Health

**Additional Editor Comments (if provided):**

**Reviewer Comments (if any, and for reference):**

Reviewer's Responses to Questions

**Comments to the Author**

1. If the authors have adequately addressed your comments raised in a previous round of review and you feel that this manuscript is now acceptable for publication, you may indicate that here to bypass the “Comments to the Author” section, enter your conflict of interest statement in the “Confidential to Editor” section, and submit your "Accept" recommendation.

Reviewer #1: All comments have been addressed

Reviewer #2: (No Response)

Reviewer #3: All comments have been addressed

2. Does this manuscript meet PLOS Digital Health’s publication criteria? Is the manuscript technically sound, and do the data support the conclusions? The manuscript must describe methodologically and ethically rigorous research with conclusions that are appropriately drawn based on the data presented.

Reviewer #1: Yes

Reviewer #2: No

Reviewer #3: Yes

3. Has the statistical analysis been performed appropriately and rigorously?

Reviewer #1: Yes

Reviewer #2: I don't know

Reviewer #3: I don't know

4. Have the authors made all data underlying the findings in their manuscript fully available (please refer to the Data Availability Statement at the start of the manuscript PDF file)?

Reviewer #1: Yes

Reviewer #2: Yes

Reviewer #3: Yes

5. Is the manuscript presented in an intelligible fashion and written in standard English?

Reviewer #1: Yes

Reviewer #2: No

Reviewer #3: Yes

6. Review Comments to the Author

Reviewer #1: Authors have addressed my comments.

Reviewer #2: I believe it may be beneficial to consider postponing the publication of this paper until the interview phase is complete. By integrating both the quantitative survey results and the qualitative interview findings into a single mixed-methods study, you can provide a more comprehensive analysis of the community’s perspectives. This approach would not only enhance the depth of your findings but also help avoid issues related to salami slicing in publication and dissemination of knowledge.

Thank you for your response regarding the comparison with previous studies. I have reviewed the revised manuscript, and it appears that the comparison with the literature you mentioned is not adequately addressed. While it is not mandatory to use the specific studies I suggested if they are not directly relevant, the key point is to contextualize your findings within the broader literature on ethical considerations from the perspectives of healthcare providers. Additionally, considering studies from regions that differ significantly from the UK or US could provide valuable insights and enhance the contribution of your work. I encourage you to expand this section to better illustrate how your findings align or contrast with existing research.

The survey may not fully capture the depth of participants' understanding of BCIs, as only a small percentage reported comprehensive knowledge. While demographic data were collected, their influence on perceptions was not thoroughly explored, which could lead to oversimplified conclusions.

Although ethical implications are mentioned, there is insufficient depth in exploring participants' views on these concerns such as the notion of "playing God".

The statistical methods employed may not adequately address potential confounders or interactions between demographic variables.

Some results, particularly in tables and figures, lack clarity and would benefit from more detailed explanations in the text. Clearer organization, especially grouping related findings together, could enhance readability and coherence.

The overrepresentation of individuals with higher education and the disproportionate number of respondents identifying as non-religious may limit the applicability of your results.

I noticed several grammatical errors, including spacing issues and inconsistencies in in-text citations.

Reviewer #3: All comments addresses, I reccomend publication.

7. PLOS authors have the option to publish the peer review history of their article (what does this mean?). If published, this will include your full peer review and any attached files.

**Do you want your identity to be public for this peer review?** For information about this choice, including consent withdrawal, please see our Privacy Policy.

Reviewer #1: No

Reviewer #2: No

Reviewer #3: No
